# PROCEEDINGS A

biogeochemistry, oceanography, environmental chemistry

air–sea gas exchange, surfactant, slick, Western Pacific, Norwegian fjords

**Author for correspondence:**
Nur Ili Hamizah Mustaffa
e-mail: nur.ili.hamizah.mustaffa@uol.de; iliehamizah@gmail.com

[†]These authors contributed equally to this work.

# Global reduction of *in situ* CO₂ transfer velocity by natural surfactants in the sea-surface microlayer

Nur Ili Hamizah Mustaffa[1,2,†],
Mariana Ribas-Ribas[2,†], Hanne M. Banko-Kubis[2] and Oliver Wurl[2]

[1]Institute for Chemistry and Biology of the Marine Environment, Carl Von Ossietzky Universität Oldenburg, 26382 Wilhelmshaven, Germany
[2]Center for Marine Sensors, Institute for Chemistry and Biology of the Marine Environment, Carl von Ossietzky Universität Oldenburg, 26382 Wilhelmshaven, Germany

NIHM, 0000-0002-3815-3446; MR-R, 0000-0003-3318-5462

For decades, the effect of surfactants in the sea-surface microlayer (SML) on gas transfer velocity ($k$) has been recognized; however, it has not been quantified under natural conditions due to missing coherent data on *in situ k* of carbon dioxide (CO₂) and characterization of the SML. Moreover, a sea-surface phenomenon of wave-dampening, known as slicks, has been observed frequently in the ocean and potentially reduces the transfer of climate-relevant gases between the ocean and atmosphere. Therefore, this study aims to quantify the effect of natural surfactant and slicks on the *in situ k* of CO₂. A catamaran, Sea Surface Scanner (S³), was deployed to sample the SML and corresponding underlying water, and a drifting buoy with a floating chamber was deployed to measure the *in situ k* of CO₂. We found a significant 23% reduction of $k$ above surfactant concentrations of $200\,\mu g\,Teq\,l^{-1}$, which were common in the SML except for the Western Pacific. We conclude that an error of approximately 20% in CO₂ fluxes for the Western Pacific is induced by applying wind-based parametrization not developed in low surfactant regimes. Furthermore, we observed an additional 62% reduction in natural slicks, reducing global

$CO_2$ fluxes by 19% considering known frequency of slick coverage. From our observation, we identified surfactant concentrations with two different end-members which lead to an error in global $CO_2$ flux estimation if ignored.

## 1. Introduction

Around half the carbon dioxide ($CO_2$) produced by humans since the Industrial Revolution has been dissolved into the ocean [1]. Consequently, understanding how the ocean absorbs $CO_2$ is critical for the prediction of climate change. The sea-surface microlayer (SML) is a diffusion layer between the ocean and the atmosphere; it covers ubiquitously the ocean surface [2]. With a thickness typically between 60 and 100 µm [3,4], the SML is a primary point for the air–sea exchange of greenhouse gases (i.e. $CO_2$, methane, dimethyl sulfate), heat and particles [5]. Meanwhile, slick is a sea-surface phenomenon of wave-dampening effect by the excessive accumulation of organic matter. Slicks are frequently observed in the ocean [6] and potentially reduce the air–sea $CO_2$ exchange by 15% [7] based on data obtained from artificial monolayers. Natural SML and slicks have not been well explored in past research programmes that estimate the fluxes of $CO_2$ into and out of the ocean. However, known biases of 20–50% in theoretical approaches [8,9], controlled tank [10–12] and field experiments involving artificial SMLs [13] justify observation under natural conditions.

For many years, parametrization of the gas transfer velocity ($k$) has involved quadratic and cubic relationships with wind speeds at a 10 m height ($U_{10}$) [14,15]. Nevertheless, several other factors affecting $k$ have been recognized, such as bubble entrainment, microbreaking, atmosphere stability, rain, fetch and the presence of surfactants, as reviewed by Wanninkhof *et al.* [14]. Surfactants or surface-active substances are a complex mixture of organic molecules that range widely in solubility and their presence in the marine environment is often biologically derived [16]. Early work by Broecker *et al.* [12] has shown that a significant relationship between $k$ and wind speed is not likely to exist for natural waters, where surfactants are influential, since the accumulation of natural surfactant in the SML forms a diffusion layer that reduces $k$ [12,13,17]. Turbulent transport in the atmosphere and water decays toward the SML. At the diffusion layer, the dominant transport process is molecular diffusion, which is the limiting step in the air–sea gas exchange. The results from theoretical and model approaches [8], as well as laboratory [18] and artificial surfactant films in field experiments [13], provide evidence that surfactant films suppress gas transfer velocities by up to 30%. A reduction of 55% has even been observed at wind speeds of 6.9–7.6 m s$^{-1}$ [13], and the wind speed range is close to average wind speed over the global ocean, i.e. 6.6 m s$^{-1}$ [19]. Besides, observation by Pereira *et al.* [20] reported the suppression of the *ex situ* $k$ of methane, which varied between 14% and 51%. This highlights a strong spatio-temporal gradient of the $k$ due to variable surfactant concentrations in the coastal waters. More recently, using sea-surface temperature (SST) as a proxy for surfactant in the SML, Pereira *et al.* [21] modelled the global reduction of $k$ by 2–32%. Such reduction is of global relevance, as a study by Wurl *et al.* [2] reported that the ocean is ubiquitously covered by the SML.

Despite the importance of $k$ parametrization in the estimation of the global uptake of climate-relevant gases by the ocean, no data exist, to the best of our knowledge, on *in situ* measurements of air–sea gas exchanges and natural surfactants in the SML. The lack of *in situ* data consequently leads to uncertainties in $k$ parametrizations, and therefore on the estimate of the oceanic $CO_2$ uptake [22]. For this reason, we provide the first *in situ* assessment of $k$ reduction by natural surfactant to investigate the effect of natural surfactant in the SML and slicks at various geographical locations and wind regimes on the $k$ of $CO_2$. Together with deployment of the catamaran Sea Surface Scanner (S$^3$) [23] to sample the SML and underlying water (ULW), a drifting buoy with a state-of-the-art floating chamber was deployed to measure the *in situ* $k$ of $CO_2$ including monitoring and correction of any potential biases from the chamber itself [24]. The field measurements have been taken in the North Atlantic Ocean, Western Pacific, Timor Sea

(offshore) and Norwegian fjords. Overall, our study ultimately leads to an understanding of how surfactants at the sea surface affect gas exchange processes under natural conditions.

## 2. Method

### (a) Field study

We collected *in situ* data during cruise FK161010 (R/V Falkor, 10 October–8 November 2016) in the Timor Sea and Western Pacific (electronic supplementary material, figure S1a), and during cruise HE491 (R/V Heincke, 8–28 July 2017) in the North Atlantic and Norwegian fjords (electronic supplementary material, figure S1b).

### (b) Biogeochemical and meteorological parameters

SML samples ($n = 89$) with a thickness of approximately 80 µm were collected using six rotating glass discs (diameter 60 cm and thickness 0.8 cm) mounted on a remote-controlled research catamaran ($S^3$) [23]. The glass discs were submerged approximately 15 cm into the water and rotated with a rate of seven rotations per minute. The SML adhered to the discs through the phenomenon of surface tension on the ascending side and was wiped off by a set of polycarbonate wipers mounted on the descending side between the discs. The thicknesses of the collected SML were in line with the SML thicknesses of $50 \pm 10$ µm using pH microelectrodes [4]. The ULW samples, taken at a depth of 1 m, were pumped simultaneously using polypropylene tubing. Discrete water samples were collected on demand from the pilot and stored onboard the $S^3$ in high-density polyethylene (HDPE) bottles in an insulated water collector (Model 6710, Teledyne ISCO, Inc., USA) at approximately 8°C. Upon recovery, all discrete samples were stored at 4°C in brown HDPE bottles prior to analysis. Meteorological data, including wind speed, were recorded at 1 min intervals using a Vintage Pro2 weather station (Davis Instruments, USA). Additional details of the sampling technique and *in situ* measurements have been reported elsewhere [23].

### (c) Measurements of the $CO_2$ transfer velocity ($k_{660}$)

During the deployment of $S^3$, an autonomous drifting buoy [24] was deployed to measure partial pressure of $CO_2$ ($pCO_2$) in the air, at a water depth of 1.2 m, and inside a floating chamber. $pCO_2$ was determined using an infrared gas analyser (OceanPack™ LI-COR LI-840x, SubCtech GmbH, Germany; range: 0–3000 µatm $\pm 1.5\%$). Aqueous $pCO_2$ was measured for 40 min, followed by two measurements taken in the floating chamber for 15 min. Air in the floating chamber was completely replaced with ambient air before each measurement. Floating chambers are the only existing technique for short-scaled spatial and temporal assessment of air–sea gas fluxes [25,26], i.e. within minutes and a few square metres, required in this study. In comparison to other indirect techniques (i.e. eddy covariance and dual tracer technique), floating chambers measure the build-up or loss of gas inside the chamber floating upside down, and thus is a direct technique to measure gas transfer. However, simple chambers often have been criticized because either the chamber protects the water surface from wind stress [27] or in very calm water bodies the chamber itself creates turbulence near the sea surface artificially enhancing the transfer velocity ($k$) [28]. The latter is unlikely to occur in the open ocean due to the presence of surface currents. To compensate the interference of the chamber by shielding the water surface from wind stress, we enhanced the floating chamber technique by measuring and comparing turbulent kinetic energy directly under and outside of the chamber's perimeter. In previous studies, we found that there is no need to correct flux data, because near-surface turbulence was not affected by the chamber [10,29]. This is most likely due to the small and shallow design of the chamber, and long fetch in oceanic environments compared to the chamber's earlier and mostly exclusive applications and assessments in lakes and estuaries: except for the application in an oceanic environment by Calleja *et al.* [26]. Despite the advantage of not inferring with the water surface directly,

other techniques (i.e. eddy covariance and dual tracer technique) do not allow for short-scaled assessments. In addition, eddy covariance requires applications of several corrections [30], and ship-based measurements are potentially error-prone due to complex motion of the ship and salt contamination of the sensors [31] as well as distortion of air flow by the ship's structure [30]. In addition, ship-based measurements using eddy covariance to measure over slicks is not possible as the ship would interfere with the integrity of the slick under observation. The dual tracer technique [32] allows assessments at stormy seas, but requires the release of large amounts of the greenhouse gas sulfur hexafluoride to the surface ocean as a tracer, and for the ship to follow the plume for several days not allowing the extensive collection of SML required in this study. During calm sea states, the plume needs to be tracked for several days exceeding the existence of slicks and requiring expensive sea time. Despite recent advances in eddy covariance, we found the chamber technique as the only applicable technique for our study to investigate small-scale variations of $CO_2$ air–sea transfer including a comparison between slick and non-slick areas.

During the FK161010 cruise, air measurements were taken for two minutes following every floating chamber cycle. During the HE491 cruise, atmospheric $pCO_2$ was measured before and after deployment on the ship's deck for approximately one hour. An air value for the whole cruise was then calculated by averaging all stable air measurements. The $CO_2$ fluxes were calculated using the following equation:

$$F_{CO_2} = \frac{dpCO_2}{dt} \frac{V}{STR},$$

where $dpCO_2/dt$ is the slope of the $pCO_2$ change in the floating chamber, $V$ is the volume of the floating chamber, $S$ represents the surface area of the floating chamber, $T$ represents the water temperature at a depth of one metre from $S^3$ and $R$ is the gas constant [24]. A positive value for $F_{CO_2}$ indicates an oceanic uptake of $CO_2$, while negative fluxes indicate a release. Measurements were excluded when the regression for the slope was $R^2 < 0.90$. The equation of the gas transfer velocity $k_w$ is

$$k_w = \frac{F_{CO_2}}{K(pCO_{2\,water} - pCO_{2\,air})}.$$

The solubility coefficient $K$ depends on the temperature and the salinity of the seawater and was calculated according to Weiss [33]. Finally, $k_w$ was standardized to $k_{660}$ with the following formula:

$$k_{660} = k_w \left(\frac{660}{Sc_{CO_2}}\right)^{-n_{Sc}},$$

where $Sc_{CO_2}$ is the temperature-dependent Schmidt number [15]. The Schmidt number exponent ($n_{Sc}$) depends on the wind speed. For low wind speeds of less than $3.7\,m\,s^{-1}$, we used $n_{Sc} = 2/3$ and for higher wind speeds, we adjusted $n_{Sc} = 1/2$ [34]. Details of $k_{660}$ calculation were published in Ribas-Ribas *et al.* [24].

## (d) Surfactant analysis

The concentration of surfactants in the SML and ULW was measured by alternating voltammetry using a VA Stand 747 (Metrohm, Switzerland) with a hanging drop mercury electrode [35]. Unfiltered samples (10 ml) were measured three to four times using a standard addition technique, where non-ionic surfactant Triton X-100 (Sigma Aldrich, Germany) was used as a standard. Concentration of surfactant is expressed as the equivalent concentration of the additional Triton X-100 ($\mu g\,Teq\,l^{-1}$). The relative standard deviations of our measurement are below 6%.

## (e) Statistical analyses

Statistical analysis was performed with R v. 3.5.3 [36] and GraphPad PRISM v. 5.0. As a further quality control analysis for floating chamber technique, we only used data with wind

speed lower than $7\,\mathrm{m\,s^{-1}}$ (to avoid breaking waves interference with the chamber). The wind speeds were grouped into low ($0$–$2.5\,\mathrm{m\,s^{-1}}$), moderate ($2.5$–$5\,\mathrm{m\,s^{-1}}$) and high regimes ($5$–$7\,\mathrm{m\,s^{-1}}$), according to Pierson & Moskowitz [37]. Surfactant concentrations were grouped into low ($50$–$200\,\mathrm{\mu g\,Teq\,l^{-1}}$), moderate ($200$–$400\,\mathrm{\mu g\,Teq\,l^{-1}}$) and high regimes ($400$–$650\,\mathrm{\mu g\,Teq\,l^{-1}}$), and slicks (greater than $1000\,\mathrm{\mu g\,Teq\,l^{-1}}$). Non-parametric tests were performed to determine whether the $k_{660}$ of $CO_2$ differed significantly between wind regimes, surfactant regimes, and sampling regions (i.e. North Atlantic, Western Pacific, offshore and Norwegian fjords). Differences were considered to be significant when $p \leq 0.05$ with a 95% confidence level. All results were reported as average $\pm$ s.d. or otherwise as indicated.

# 3. Results and discussion

## (a) Parametrization of gas transfer velocity in the field measurements

In general, *in situ* $k_{660}$ ranged between $1.5\,\mathrm{cm\,h^{-1}}$ and $85.1\,\mathrm{cm\,h^{-1}}$ and increased with $U_{10}$ observed in a range of $0.4$–$7.0\,\mathrm{m\,s^{-1}}$ (figure 1). The regression of our data ($k_{660} = 9.4\,(\pm4.9) + 0.6\,(\pm0.1)\,{}^*U_{10}{}^2$) was higher than the range of existing parametrizations ($k_{660} = 0.25\,{}^*U_{10}{}^2$ to $k_{660} = 0.39\,{}^*U_{10}{}^2$) [15,38,39]. The trend of our $k_{660}$ with $U_{10}$ was similar to field data from Donelan & Drennan [40] (see fig. 3 in [40]). However, our $k_{660}$'s were lower than those predicted [40], where a large variance in $k_{660}$ between field measurements and laboratory was reported. The same authors proposed that increased wave dissipation in the field provides a source of turbulence in the near-surface that acts to reduce the resistance of air–sea $CO_2$ transfer. We suggest that the high $k_{660}$ in the our study, exclusively observed in the Western Pacific (see electronic supplementary material, figure S2), is due to the low resistance of air–sea $CO_2$ transfer by the lowest concentration of surfactants observed on a global scale [2,41]. A similar weak wind dependence was also observed by McGillis *et al.* [42], who suggested that other factors, such as incidental solar radiation, phytoplankton biomass and surface ocean stratification, can have significant effect on air–sea gas exchange. Overall, wind speed as a proxy for near-surface turbulence cannot fully explain the $k_{660}$; this is especially true with the increasing influence of buoyancy fluxes at lower wind regimes [42], which strengthen the potential influence of surfactant in air–sea gas transfer parametrizations [12,22].

## (b) Reduction of gas transfer velocity by surfactants

The ranges of observed $k_{660}$ within three surfactant regimes and wind regimes are presented in figure 2a,b, respectively. The average of $k_{660}$ showed a decreasing trend with increasing surfactant concentrations; it decreased by 73%, from a low to a moderate surfactant regime (figure 2a). The highest average $k_{660}$ observed within the low surfactant regime ($33.4 \pm 37.8\,\mathrm{cm\,h^{-1}}$, $n = 147$) was significantly different from the other regimes (Kruskal–Wallis with Dunn's multiple comparison $p < 0.0001$). It is particularly true for the very high $k_{660}$ observed in the Western Pacific due to the missing resistance in this low-surfactant regime compared to the other oceanic regions [2]. In addition, the average $k_{660}$ values were significantly lower within the low wind regime ($6.4 \pm 7.4\,\mathrm{cm\,h^{-1}}$, $n = 50$, figure 2b), compared to moderate ($24.6 \pm 21.8\,\mathrm{cm\,h^{-1}}$, $n = 52$) or high wind regimes ($24.0 \pm 23.3\,\mathrm{cm\,h^{-1}}$, $n = 46$) ($p < 0.0001$). The differences of $k_{660}$ between the moderate and high surfactant regimes, as well as between moderate and high wind regimes, were insignificant ($p > 0.05$). Overall, wind speed and surfactant were indeed found to affect the $k_{660}$ parametrization by enhancing [15,43] and reducing [13,21] the $k_{660}$, respectively; however, the interactive effect between wind speed and surfactant was insignificant (two-way analysis of variance; $p = 0.701$), indicating an uncoupled effect. This is because high wind speed functions both ways: initially, the integrity of the SML is disturbed through breaking waves with the consequence of increased $k_{660}$. However, after breaking through the SML, waves enhance the reformation of the organically enriched SML through ascending bubbles from the water column [44].

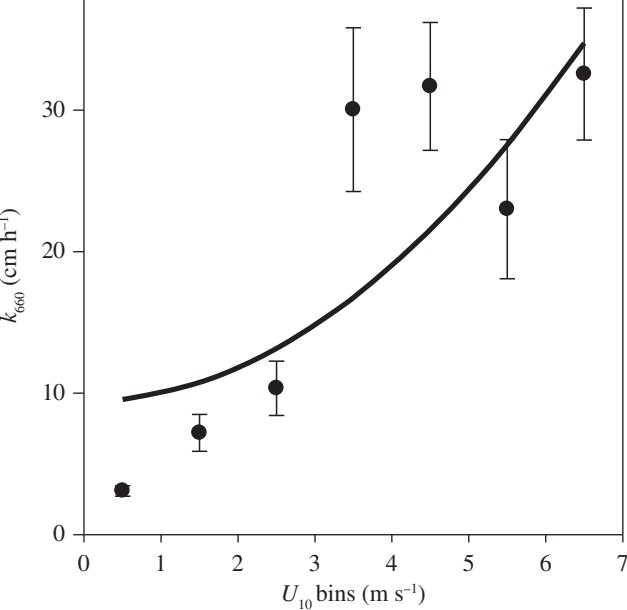

**Figure 1.** Distribution of *in situ* $k_{660}$ versus wind bins ($U_{10}$). The black dots are the average of $k$ for each wind bin and the error bars are the standard error of the average for each wind bin. Quadratic regression of our study; $k_{660} = 9.4\,(\pm 4.9) + 0.6\,(\pm 0.1)\,{}^{*}U_{10}{}^{2}$.

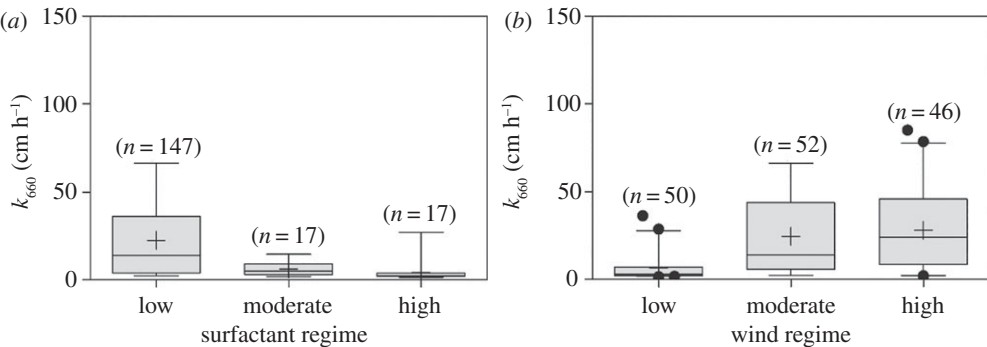

**Figure 2.** Whisker box plot of *in situ* $k_{660}$. (*a*) At different surfactant regimes and (*b*) at different wind regimes. Error bars represent 5–95% of median values. Cross symbols represent mean values, lines represent median values and black points represent the outliers. *n*, number of observations.

Our *in situ* data show that $k_{660}$ is reduced by 23% at surfactant concentrations exceeding 200 µg Teq l$^{-1}$ (figure 3*a*). We further observed that the trend of $k_{660}$ reduction is similar by considering surfactant concentrations either in the SML (figure 3*a*) or in the ULW (electronic supplementary material, figure S3a). Moreover, a significant correlation was found between surfactants in the SML and ULW ($R^2 = 0.921$, $p < 2.2 \times 10^{-16}$, $n = 84$), indicating consistent enrichment processes of the SML [20]. While we applied the surfactant concentrations of the SML to correlate $k_{660}$ (figure 3*a*), we propose that surfactant concentrations in either the SML or the ULW can be used to parametrize $k_{660}$. This provides a new perspective in improving parametrizations, as logistically challenging SML sampling is not required. Overall, the surfactant concentration in the SML ranged from 52 to 4760 µg Teq l$^{-1}$ ($n = 94$, figure 3*a,b*, respectively).

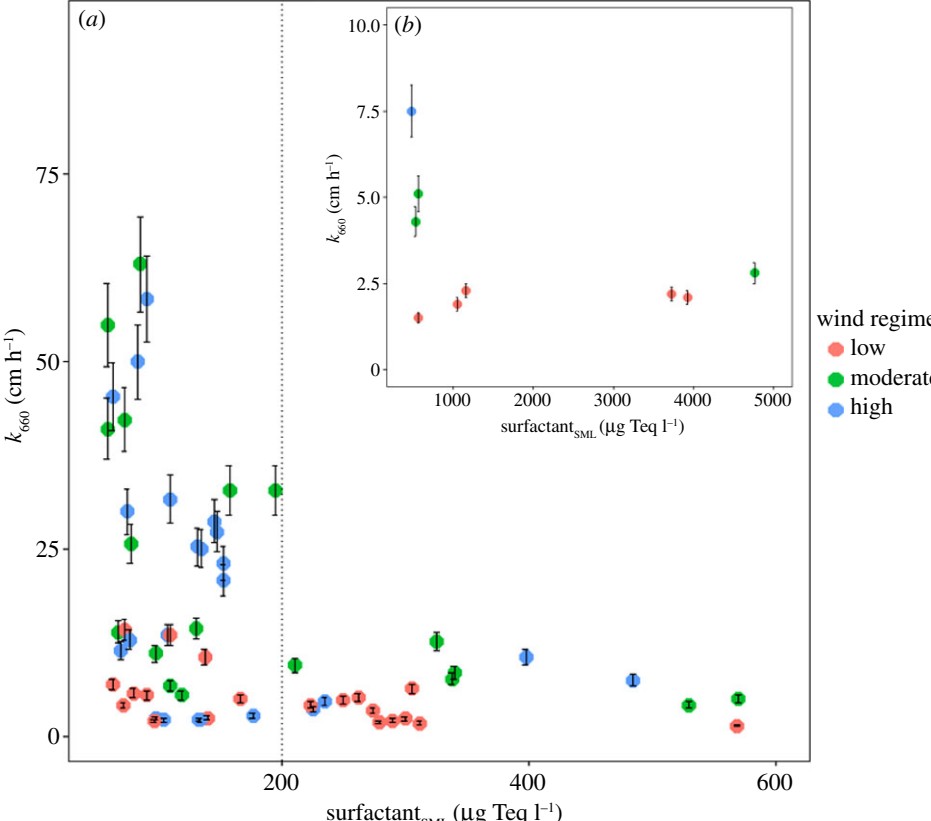

**Figure 3.** Scattered plots of $k_{660}$ and surfactant concentrations in the SML. (*a*) $k_{660}$ reduced by natural surfactant in the SML. The vertical line at 200 µg Teq l$^{-1}$ indicates a breaking point of surfactant in the SML. (*b*) $k_{660}$ reduced by surfactant during intensive slick of cyanobacteria bloom. Colour plot represents wind regimes. Error bars represent 10% of standard error of $k_{660}$. (Online version in colour.)

The dotted vertical line represents the breakpoint of surfactant (i.e. 200 µg Teq l$^{-1}$) calculated with piecewise analysis. Below the breakpoint (less than 200 µg Teq l$^{-1}$), we found the lowest surfactant concentrations in the Western Pacific, one of the most oligotrophic regions of the global ocean [2]. However, surfactant concentrations above the breakpoint (greater than 200 µg Teq l$^{-1}$) were found in other oligotrophic regions, including in the North Pacific, subtropical North Pacific and Arctic Oceans [2]. The range of $k_{660}$ measured at surfactant <200 µg Teq l$^{-1}$ scattered widely between 2.1 and 63.0 cm h$^{-1}$ (20.3 ± 17.2 cm h$^{-1}$, $n = 41$), and a regression was insignificant; $k_{660} = -0.10 \, (\pm 0.07) \times$ surfactant <200 µg Teq l$^{-1}$ ($R^2 = 0.024$, $p = 0.165$, $n = 39$). Surprisingly, below the surfactant breakpoint (less than 200 µg Teq l$^{-1}$), we observed the lowest $k_{660}$ at high wind speeds. We suggest that a reduction of $k_{660}$ below the surfactant breakpoint might be masked by opposing processes (e.g. near-surface stratification or primary production). However, no significant linear regression between $k_{660}$, surfactant less than 200 µg Teq l$^{-1}$, and solar radiation was found ($R^2 = 0.092$, $p = 0.015$, $n = 61$). A reduction of $k$ by 7% at a surfactant concentration of 155 µg Teq l$^{-1}$ was reported by Pereira *et al.* [21], but a low number of samples limited their conclusion. Additionally, by comparing the averaged $k_{660}$ values between surfactant concentrations less than 200 µg Teq l$^{-1}$ and those greater than 200 µg Teq l$^{-1}$, we found that the average of $k_{660}$ decreased by 78% with less scattering when surfactant concentrations were greater than 200 µg Teq l$^{-1}$. The $k_{660}$ ranged between 1.5 and 12.7 cm h$^{-1}$ (5.4 ± 3.1 cm h$^{-1}$, $n = 20$), although 50% of the measurement was taken at moderate to high wind speeds. This

observation is in line with Salter *et al.* [13], who found a reduction of $k$ by up to 55% even at higher wind speeds (6.9–7.6 m s$^{-1}$) in the presence of an artificial surfactant film. By excluding extreme surfactant concentrations in intensive slicks (figure 3$b$), our surfactant concentrations above the breakpoint (greater than 200 μg Teq l$^{-1}$) were in a similar range measured in the North Atlantic (201–669 μg Teq l$^{-1}$) [41] with a reduction of $k_w$ between 7 and 32%, measured *ex situ* [21] (compared to 23% in our study). Similar to Pereira *et al.* [21], we observed no linear regression between $k_{660}$ and surfactant greater than 200 μg Teq l$^{-1}$ ($R^2 = -0.055$, $p = 0.975$, $n = 18$). However, our results show clearly that the presence of surfactants in the top 1 m layer affects $k$.

## (c) Reduction of gas transfer velocity by slick

Slick is a surface phenomenon that has a wave-dampening effect due to substantial accumulation of surfactants in the SML. It has a proposed threshold value of greater than or equal to 1000 μg Teq l$^{-1}$ [45]. During the cruise in the Western Pacific (FK161010), we observed large slicks formed by an intensive surface bloom of cyanobacteria (*Trichodesmium* sp.) at Station 4 [46], i.e. within our defined offshore regime. A key finding of our study is that slicks reduce $k_{660}$ by 62% (figure 4$a$), probably due to the presence of a thicker diffusion layer of surfactants and microbial metabolism [7]. For example, the SML from slicks shows the highest average surfactant concentration (figure 4$b$), i.e. 2925 ± 1704 μg Teq l$^{-1}$ ($n = 5$) at Station 4 (surface bloom). The surfactant concentrations at non-slick Stations 5B (offshore regime), 16 and 17 (both open ocean) were 120 ± 58 μg Teq l$^{-1}$ ($n = 5$), 66 ± 13 μg Teq l$^{-1}$ ($n = 4$) and 86 ± 12 μg Teq l$^{-1}$ ($n = 4$), respectively. We observed the lowest average of $k_{660} = 2.2 ± 0.4$ cm h$^{-1}$ ($n = 12$) at Station 4. The difference was insignificant (Kruskal–Wallis with Dunn's multiple comparison, $p > 0.05$) compared to Station 5A, which was influenced by slicks without the presence of the bloom ($k_{660} = 2.5 ± 0.3$ cm h$^{-1}$, $n = 10$). Stations 5A and 5B (which had the same starting position of drift) were located 51 nautical miles north of Station 4. Station 5B (no slicks) exhibited a significantly higher $k_{660}$, with an average of 12.0 ± 2.2 cm h$^{-1}$ ($n = 9$) ($p < 0.001$) compared to slick-influenced Stations 4 and 5A; however, all observations were made within a narrow range of wind speed ($U_{10} = 0.5$–3.7 m s$^{-1}$). Additionally, the averages of $k_{660}$ between Stations 5A and 5B were significantly different ($p = 0.003$), but surfactant concentrations at Station 5A were not available to further explain the influence of slick on $k_{660}$. Earlier work by Frew [22] demonstrated that $k$ may be suppressed by surfactant even at low wind speed. The effect of surfactants at low wind speed is associated with microscale wave breaking through their individual contributions to mean square wave slope, which is observed to correlate with $k$ [22]. The lower $k_{660}$ at Station 5A, compared to Station 5B without the presence of slicks, is mainly reduced by the slick characteristics, i.e. high concentrations of surfactants, which causes dampening of capillary waves and, therefore, the mean square wave slope. Stations 16 and 17 were both located in the Western Pacific regime; average $k_{660}$ were 56.8 ± 9.8 cm h$^{-1}$ ($n = 7$) and 33.1 ± 13.9 cm h$^{-1}$ ($n = 8$), respectively. To our knowledge, only a few field studies have explored the effect of artificial slicks on $k_{660}$. For example, the field measurements using artificial slicks of oleyl alcohol, reducing micro-scaled turbulence under the surface by dampening capillary waves, indicated suppression of $k_{660}$ up to 30% and 55% at low (1.5–3.0 m s$^{-1}$) [47] and high wind speeds (6.9–7.6 m s$^{-1}$) [13], respectively. However, the insoluble properties of oleyl alcohol form a monolayer film that does not completely simulate a natural slick with its biofilm-like [7] and rheological properties [48], the latter through increased thickness (compared to non-slick SML or monolayers), and the presence of complex mixtures of soluble and insoluble surfactants [49]. Our recent study [46] showed that the patches of intensive slicks (which therefore have increased thickness) of cyanobacteria (*Trichodesmium* sp.) provided an additional barrier on the SML, reducing heat exchange and evaporation rates. Similarly, it explained the lower $k_{660}$ values at slick stations (Stations 4 and 5A).

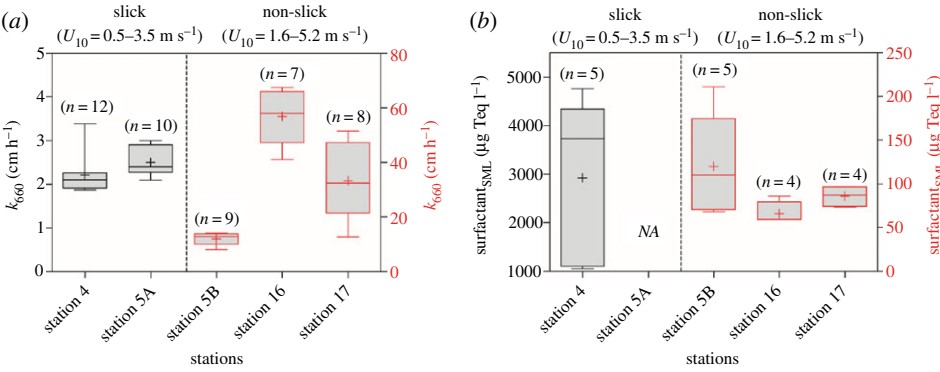

**Figure 4.** Whisker box plot of $k_{660}$ during cruise in the Western Pacific (FK161010). Box plots of (a) $k_{660}$ and (b) surfactant concentrations in the SML at selected stations. Error bars represent 5–95% median values. Lines represent 50% median and cross symbols represent mean values. $n$, number of observations. (Online version in colour.)

## (d) The gas transfer velocity at different oceanic regimes

Figure 5 shows the average surfactant concentrations in the SML grouped into four oceanic regimes, specifically, the open North Atlantic and Western Pacific, offshore (Timor Sea; within a distance of approx. 10 km to the shoreline) and the Norwegian fjords. The results show that a significant relationship between $k_{660}$, wind speed and surfactant is dependent on the geographical location. The measurements of $k_{660}$ in the North Atlantic (figure 5a) and Western Pacific (figure 5b) were made during moderate to high wind regimes with an average $k_{660}$ of $38.9 \pm 10.1$ cm h$^{-1}$ ($n = 17$) and $47.9 \pm 18.8$ cm h$^{-1}$ ($n = 35$), respectively. The average surfactant concentrations in the North Atlantic and Western Pacific were $139 \pm 36$ µg Teq l$^{-1}$ ($n = 8$) and $82 \pm 21$ µg Teq l$^{-1}$ ($n = 13$), respectively. The measurements in the offshore regime (figure 5c) were taken during low to high wind regimes and average $k_{660}$ was $5.9 \pm 4.3$ cm h$^{-1}$ ($n = 46$). No significant trend was observed between $k_{660}$ and surfactant in the offshore regime ($114 \pm 47$ µg Teq l$^{-1}$, $n = 23$) (figure 5c). The fjord regime (figure 5d) exhibited an average $k_{660}$ of $9.2 \pm 7.2$ cm h$^{-1}$ ($n = 39$), with the highest average surfactant of $316 \pm 131$ µg Teq l$^{-1}$ ($n = 21$). A clear decreasing trend of $k_{660}$ and a significant correlation were found (figure 5d), where $k_{660} = 2.13$ ($\pm 0.49$) *$U_{10}$ − 0.02 ($\pm 0.01$) *surfactant$_{SML}$ ($R^2 = 0.546$, $p = 0.0003$, $n = 18$). Our average $k_{660}$ for the fjords and offshore regimes were in the same range as other coastal studies (6.8–22.1 cm h$^{-1}$) [20,24] and suppression of $k$ was five times higher in the coastal water compared to oceanic water [12]; i.e. $k$ was higher in the oceanic water.

Spatio-temporal variations in $k_{660}$ are affected by different geographical and biological regimes, which are in turn influenced by physical forces, such as bubble and wave spectra, wind speed, whitecap fraction and tidal currents [50]. An early study by Lee *et al.* [51] demonstrated that hydrodynamic effects on air–sea gas exchange depend on surfactant type. For example, high-molecular-weight surfactants with protein structures are more effective in reducing the $k$ compared to extremely soluble ionic surfactants [51]. Phytoplankton exudes surfactants [16] and is found throughout the SML [52]. Hood *et al.* [53] showed that dissolved organic matter in the fjord water influenced by glacial melt is associated with labile proteinaceous material and is less aromatic. In addition, different compositions of surfactants with distance from the coast (along a 20 km transect) led to higher variability (between 14% and 51%) of $k$ in the coastal regime [20]. For instance, Frew [22] demonstrated that the relationship between $k$ and wind speed was bilinear (see fig. 5.2 in Frew [22]) and varied in space and time due to changes in the composition of organic matter in the ULW. Our earlier study showed that longer residence time and prolonged exposure to solar radiation enhanced photodegradation of organic matter in the SML [54], leading to lower surfactant concentrations in the open ocean (i.e. the Western Pacific and North Atlantic regimes). Lower concentrations and potentially more recalcitrant surfactants in the Western Pacific and

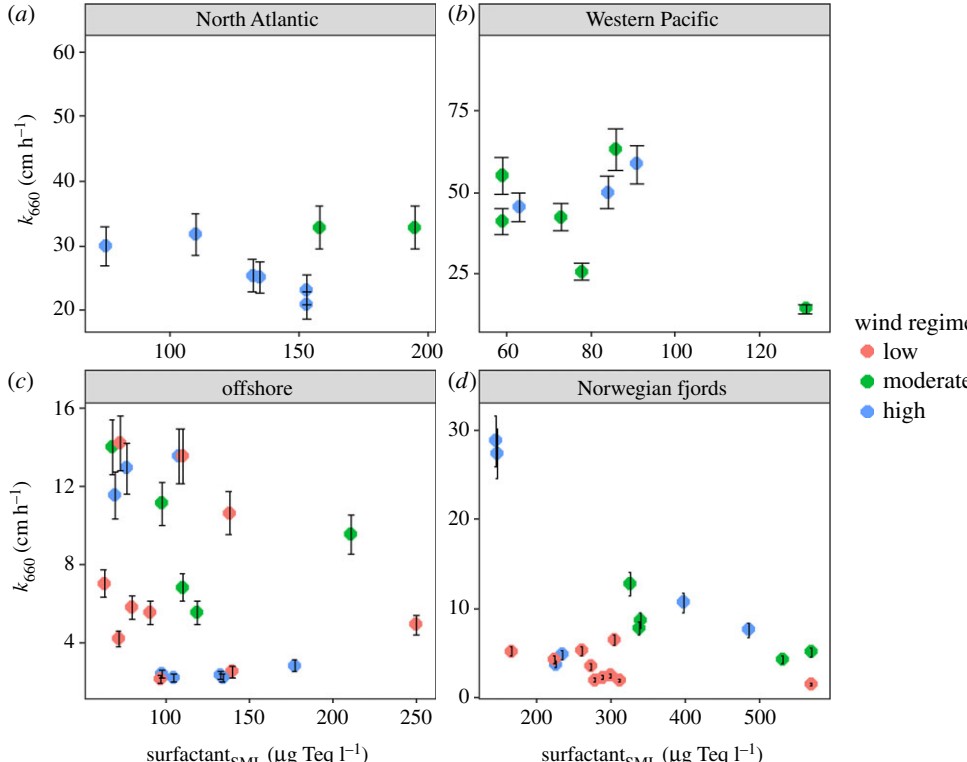

**Figure 5.** Scattered plots of $k_{660}$ and surfactant concentrations in the SML at different oceanic regimes. (*a*) North Atlantic, (*b*) Western Pacific, (*c*) offshore and (*d*) Norwegian fjords regimes. Colour plot represents wind regimes. Error bars represent 10% of standard error of $k_{660}$. (Online version in colour.)

North Atlantic regimes showed less significant effects on $k_{660}$ reduction (figure 5*a,d*, respectively) compared to the potentially more labile surfactants in the fjord (figure 5*b*). Moreover, the lower $k_{660}$ values (less than $20 \, \mathrm{cm \, h^{-1}}$) observed in all regimes were potentially influenced by other physical forces with effects on patchy features of the SML on spatial and temporal scales of less than 50 m and a few minutes, respectively [55].

Parametrization of gas exchange in the equatorial Pacific [42] showed that $CO_2$ exchange in this region was forced by buoyancy fluxes rather than solely by low wind speed. A recent study used the Atlantic Ocean SST as a spatio-temporal proxy for surfactants in the global ocean [21], where it was expected that warmer surface oceans produced more biologically derived surfactants. This is contrary to our findings; we did not find any significant relationship between SST and/or *in situ* $k_{660}$ ($R^2 = 0.009$, $p = 0.132$, $n = 133$) and surfactants ($R^2 = 0.408$, $p = 5.59 \times 10^{-9}$, $n = 63$). We also observed relatively low surfactant concentrations in the warmer and oligotrophic Western Pacific regimes (figure 5*b*) with an average SST of $30.5 \pm 0.9$°C ($n = 60$). Despite a large difference in the average SST between the Western Pacific, including offshore sites ($30.8 \pm 1.2$°C, $n = 57$), and the North Atlantic ($13.8 \pm 0.3$°C, $n = 21$) regimes, the surfactant concentrations in those regimes were similar (figure 5*a–c*, respectively). From our data, we concluded that concentration of biologically derived surfactants depends not only on SST for primary production but also on levels of nutrients and, to a lesser extent, on light regimes. In addition, terrestrial input and atmospheric deposition are also sources of surfactants for coastal water and the SML [5,56]. Our *in situ* data indicate that knowledge of surfactants in the SML supports a better understanding of the variability in parametrizations of $k_{660}$, but the complexity probably excludes an approach with a single proxy to describe $k$ satisfactorily.

## (e) Global implications in air–sea gas exchange

Our results support an early hypothesis [12] in which a robust relationship between $k$ and wind speed is not likely to exist for natural waters where surfactants are influential. During two years of observations, Wurl *et al.* [2] showed that surfactants are enriched in the SML of oligotrophic regions; they, therefore, concluded that the SML covers the ocean on a global scale. Slicks have been reported [6] to cover the ocean, with coverage of 30% and 11% in the coastal and open ocean, respectively. Our data show that the $k_{660}$ is reduced by 23% by surfactants above the observed breakpoint (i.e. 200 µg Teq l$^{-1}$) and 62% in the presence of slicks. Global observation by Takahashi *et al.* [57] reported that the Pacific ($-0.48$ Tg carbon year$^{-1}$) and Atlantic Oceans ($-0.58$ Tg carbon year$^{-1}$) are the net sink of anthropogenic $CO_2$. Using a similar approach to that of Wurl *et al.* [7], we calculated the reduction of $CO_2$ fluxes by surfactants for non-slick and slick conditions at up to 20% and 7%, respectively, in the open ocean (Western Pacific and North Atlantic; electronic supplementary material, tables S1 and S2, respectively). Meanwhile, the reduction of $CO_2$ fluxes in the Norwegian fjords was 16% during non-slick conditions and 19% for slicks (electronic supplementary material, tables S1 and S2, respectively). The percentage of reductions was close to previous estimation (i.e. 15%) [7] during slick conditions in the Mediterranean Sea. Previous assessment of $k$ parametrizations has been made in the regions with higher surfactant concentrations (i.e. close to the coastline) [15,38]. Therefore, if we applied the commonly used $k$ parametrization [15,38] in the Western Pacific, which consists of low surfactants, a bias of approximately 20% in the estimation of $CO_2$ air–sea fluxes could potentially exist. Overall, our results indicated the importance of natural surfactant, including slicks, in $k$ parametrizations and, therefore, on $CO_2$ air–sea fluxes on regional and global scales. With the technologies we developed to measure *in situ* $k$ of $CO_2$ [24] and simultaneously collect the SML [23], we enhanced the understanding of the effect of natural surfactants on $k$ parametrization.

Data accessibility. S$^3$ catamaran and pCO$_2$ data of FK161010 are archived at the PANGAEA data publisher [58,59]. S$^3$ catamaran and pCO$_2$ data of HE491 are archived at the PANGAEA data publisher [60,61].
Authors' contributions. N.I.H.M., M.R.-R. and O.W. contributed to data collection during cruise FK161010. N.I.H.M., H.M.B.-K. and O.W. contributed to data collection during cruise HE491. N.I.H.M. performed surfactant analysis. M.R.-R. and H.M.B.-K. calculated the fluxes and gas transfer velocity. N.I.H.M., M.R.-R. and O.W. drafted and revised the manuscript. All authors reviewed the manuscript before submission.
Competing interest. The authors declare there is no conflict of interest.
Funding. This research was funded by the European Research Council (ERC), grant no. GA336408, through a PASSME project.
Acknowledgements. We thank the Schmidt Ocean Institute (SOI), the captain and the crew of R/V Falkor for their assistance during the AIR ↓↑ SEA expedition (cruise no. FK161010). We thank the captain and the crew of R/V Heincke during the cruise HE491. Many thanks to all scientific crew members on board for their help and support during both cruises.

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
