## [Reviewer comments · Proceedings. Mathematical, Physical, and Engineering Sciences]

Review History

RSPA-2019-0307.R0 (Original submission)

Review form: Referee 1

Is the manuscript an original and important contribution to its field?

Good

Is the paper of sufficient general interest?

Good

Is the overall quality of the paper suitable?

Marginal

Can the paper be shortened without overall detriment to the main message?

Yes

Do you think some of the material would be more appropriate as an electronic appendix?

No

Do you have any ethical concerns with this paper?

No

Recommendation?

Major revision is needed (please make suggestions in comments)

Comments to the Author(s)

See attached file

Review form: Referee 2

Is the manuscript an original and important contribution to its field?

Good

Is the paper of sufficient general interest?

Good

Is the overall quality of the paper suitable?

Acceptable

Can the paper be shortened without overall detriment to the main message?

Yes

Do you think some of the material would be more appropriate as an electronic appendix?

No

Do you have any ethical concerns with this paper?

No

Recommendation?

Accept with minor revision (please list in comments)

Comments to the Author(s)

General Comment

The manuscript discusses the effect of surfactants in the sea-surface microlayer (SML) on gas transfer velocity $15(k)$ under natural conditions. An error of flux estimation in the western pacific is induced by applying wind-based parameterization not developed in low surfactant regimes. Reduction in natural slicks reduces the global flux of CO₂ by 19%. Overall the study is very interesting and significant. The manuscript needs to be improved before it can be accepted for publication.

Detail Comment

I think the authors need to improve their abstract. What are the main objectives of this study? Its is good if they can briefly mention the approach of their study.

The introduction is not well structured as well. The authors need to clearly mention the motivation for the study. What they already knew and what the want to cover for this study.

I suggest the authors improve their paragraphing in the Introduction.

Line 276: name the study that considers the effect of slick on K660.

Decision letter (RSPA-2019-0307.R0)

09-Sep-2019

Dear Dr Mustafa:

I am writing to inform you that your manuscript RSPA-2019-0307 entitled "Global reduction of in-situ CO₂ transfer velocity by natural surfactants in the sea surface microlayer" has been rejected in its present form for publication in Proceedings A.

The Editor has made this decision based on the advice of referees, and taking into account their own opinion of your paper. With this in mind we would like to invite a resubmission, provided the comments of the referees and any comments from the Editor are taken into account. This is not a provisional acceptance.

The resubmission will be treated as a new manuscript. Please note that resubmissions must be submitted within six months of the date of this email. In exceptional circumstances, extensions may be possible if agreed with the Editorial Office.

Please find below the comments made by the referees, not including confidential reports to the Editor, which I hope you will find useful. If you do choose to resubmit your manuscript, please include details of how you have responded to the comments, and the adjustments you have made.

Please note that we have a strict upper limit of 28 pages for each paper. Please endeavour to incorporate any revisions while keeping the paper within journal limits. Please note that page charges are made on all papers longer than 20 pages. If you cannot pay these charges you must reduce your paper to 20 pages before submitting your revision. Your paper has been ESTIMATED to be 21 pages. We cannot proceed with typesetting your paper without your agreement to meet page charges in full should the paper exceed 20 pages when typeset. If you have any questions, please do get in touch.

To upload a resubmitted manuscript, log into <http://mc.manuscriptcentral.com/prsa> and enter your Author Centre, where you will find your manuscript title listed under "Manuscripts with Decisions." Under "Actions," click on "Create a Resubmission." Please be sure to indicate that it is a resubmission, and ensure you enter this ID - RSPA-2019-0307 - as the previous submission number.

Yours sincerely
Raminder Shergill
proceedingsa@royalsociety.org

on behalf of
Professor Gregory Ivey
Board Member
Proceedings A

Reviewer(s)' Comments to Author:

Referee: 1

Comments to the Author(s)
See attached file

Referee: 2

Comments to the Author(s)

General Comment

The manuscript discusses the effect of surfactants in the sea-surface microlayer (SML) on gas transfer velocity 15 (k) under natural conditions. An error of flux estimation in the western pacific is induced by applying wind-based parameterization not developed in low surfactant regimes. Reduction in natural slicks reduces the global flux of CO_2 by 19%. Overall the study is very interesting and significant. The manuscript needs to be improved before it can be accepted for publication.

Detail Comment

I think the authors need to improve their abstract. What are the main objectives of this study? Its is good if they can briefly mention the approach of their study.

The introduction is not well structured as well. The authors need to clearly mention the motivation for the study. What they already knew and what the want to cover for this study.

I suggest the authors improve their paragraphing in the Introduction.

Line 276: name the study that considers the effect of slick on K660.

Board member pre-assessment comments (if available):

Author's Response to Decision Letter for (RSPA-2019-0307.R0)

See Appendix A.

RSPA-2019-0763.R0

Review form: Referee 1

Is the manuscript an original and important contribution to its field?

Good

Is the paper of sufficient general interest?

Good

Is the overall quality of the paper suitable?

Good

Can the paper be shortened without overall detriment to the main message?

Yes

Do you think some of the material would be more appropriate as an electronic appendix?

No

Do you have any ethical concerns with this paper?

No

Recommendation?

Accept as is

Comments to the Author(s)

I am completely unpersuaded by the response of the authors and I think they are kidding themselves if they think the data in this paper is accurate. For instance, the authors use measurements of bulk-phase turbulence inside and outside their chamber as evidence that the presence of the chamber is not affecting the surface microlayer. However, it is well known that measurements of bulk phase turbulence do not accurately represent conditions within a few hundred microns of the water surface. So, sure, turbulence a few centimeters or tens of centimeters below the surface is the same inside and outside the chamber, but the authors have no way of knowing, and have provided no information, on whether the surfactant concentration in the surface microlayer is the same. However, I can't prove the chamber is affecting the microlayer and more than the authors can prove there is no effect.

Similarly, the authors' accusation that I am cherry-picking studies to support my case that their transfer velocities are outside what might be expected is insulting. Instead, they pick a study done from the very first time direct-variance flux measurements gave anything near a sensible answer as justification for their anomalous data. The point of picking the studies I did was they spanned a range of conditions and were done by different groups, but mostly it was simply that I happened to have those datasets in digital form at hand on my hard drive for plotting. Most importantly however, I could have picked nearly *any* credible open-ocean gas transfer dataset and it would have plotted almost on top of the data shown in the figure in my review. In contrast, the authors' data is clearly more than one sigma outside the variance. Again, however, I admit I cannot prove there is an issue with the gas transfer data presented in this manuscript. Just that it looks very different from nearly every other measurement of gas transfer made in the last 15 years.

Gas exchange research will survive yet another paper published with questionable data in it. This can be the new Smith and Jones.

Review form: Referee 2**Is the manuscript an original and important contribution to its field?**

Excellent

Is the paper of sufficient general interest?

Excellent

Is the overall quality of the paper suitable?

Excellent

Can the paper be shortened without overall detriment to the main message?

Yes

Do you think some of the material would be more appropriate as an electronic appendix?

Yes

Do you have any ethical concerns with this paper?

No

Recommendation?

Accept as is

Comments to the Author(s)

The authors have improved their manuscript based on my comment before

Decision letter (RSPA-2019-0763.R0)

06-Jan-2020

Dear Dr Mustaffa

On behalf of the Editor, I am pleased to inform you that your manuscript entitled "Global reduction of in-situ CO₂ transfer velocity by natural surfactants in the sea surface microlayer" has been accepted in its final form for publication in Proceedings A.

Our Production Office will be in contact with you in due course. You can expect to receive a proof of your article soon. Please contact the office to let us know if you are likely to be away from e-mail in the near future. If you do not notify us and comments are not received within 5 days of sending the proof, we may publish the paper as it stands.

Open access

You are invited to opt for open access, our author pays publishing model. Payment of open access fees will enable your article to be made freely available via the Royal Society website as soon as it is ready for publication. For more information about open access please visit http://royalsocietypublishing.org/site/authors/open_access.xhtml. The open access fee for this journal is £1700/\$2380/€2040 per article. VAT will be charged where applicable.

Note that if you have opted for open access then payment will be required before the article is published – payment instructions will follow shortly. If you wish to opt for open access then please inform the editorial office (proceedingsa@royalsociety.org) as soon as possible.

Your article has been estimated as being 15 pages long. Our Production Office will inform you of the exact length at the proof stage.

Proceedings A levies charges for articles which exceed 20 printed pages. (based upon approximately 540 words or 2 figures per page). Articles exceeding this limit will incur page charges of £150 per page or part page, plus VAT (where applicable).

Under the terms of our licence to publish you may post the author generated postprint (ie. your accepted version not the final typeset version) of your manuscript at any time and this can be made freely available. Postprints can be deposited on a personal or institutional website, or a recognised server/repository. Please note however, that the reporting of postprints is subject to a media embargo, and that the status the manuscript should be made clear. Upon publication of the definitive version on the publisher's site, full details and a link should be added.

You can cite the article in advance of publication using its DOI. The DOI will take the form: 10.1098/rspa.XXXX.YYYY, where XXXX and YYYY are the last 8 digits of your manuscript

number (eg. if your manuscript number is RSPA-2017-1234 the DOI would be 10.1098/rspa.2017.1234).

For tips on promoting your accepted paper see our blog post:
<https://blogs.royalsociety.org/publishing/promoting-your-latest-paper-and-tracking-your-results/>

Thank you for your submission. On behalf of the Editors of the journal, we look forward to your continued contributions to the Journal.

Best wishes
Raminder Shergill,
Proceedings A Editorial Office
proceedingsa@royalsociety.org

on behalf of
Professor Gregory Ivey
Board Member
Proceedings A

Reviewer(s)' Comments to Author:

Referee: 2

Comments to the Author(s)
The authors have improved their manuscript based on my comment before

Referee: 1

Comments to the Author(s)
I am completely unpersuaded by the response of the authors and I think they are kidding themselves if they think the data in this paper is accurate. For instance, the authors use measurements of bulk-phase turbulence inside and outside their chamber as evidence that the presence of the chamber is not affecting the surface microlayer. However, it is well known that measurements of bulk phase turbulence do not accurately represent conditions within a few hundred microns of the water surface. So, sure, turbulence a few centimeters or tens of centimeters below the surface is the same inside and outside the chamber, but the authors have no way of knowing, and have provided no information, on whether the surfactant concentration in the surface microlayer is the same. However, I can't prove the chamber is affecting the microlayer and more than the authors can prove there is no effect.

Similarly, the authors' accusation that I am cherry-picking studies to support my case that their transfer velocities are outside what might be expected is insulting. Instead, they pick a study done from the very first time direct-variance flux measurements gave anything near a sensible answer as justification for their anomalous data. The point of picking the studies I did was they spanned a range of conditions and were done by different groups, but mostly it was simply that I happened to have those datasets in digital form at hand on my hard drive for plotting. Most importantly however, I could have picked nearly *any* credible open-ocean gas transfer dataset and it would have plotted almost on top of the data shown in the figure in my review. In contrast, the authors' data is clearly more than one sigma outside the variance. Again, however, I admit I cannot prove there is an issue with the gas transfer data presented in this manuscript. Just that it looks very different from nearly every other measurement of gas transfer made in the last 15 years.

Gas exchange research will survive yet another paper published with questionable data in it. This can be the new Smith and Jones.

Appendix A

Reviewer 1

Comments	Response
This is an interesting paper but I have several main issues with it. Primarily I do not think the authors have sufficiently demonstrated the efficacy of the floating-dome method for this particular application. The main issue, in my opinion, is that when you seal a small area of ocean surface underneath the dome and isolate it from the wind stress, it could allow formation of a more stable and enriched surface microlayer. The situation would be somewhat similar to stirred-tank or wind tunnel experiments, which were found to be sensitive to contamination from surfactants since the surfactants accumulate on the surface and are never mixed into the bulk. Going back through the literature on headspace measurements, I do not see a study where anyone has used a headspace chamber in the laboratory in a wind tunnel with and without surfactants. If such an experiment were conducted it would provide evidence that sealing the surface from the wind stress is not affecting the surface chemistry of the water. Then it could be concluded that the gas flux inside the chamber is the same as the gas flux through the water surface (such a demonstration is certainly not in the references 23 and 24 from the manuscript).	We thank for reviewer comment. We would like to emphasize that we do not use a simple chamber/dome, but an advanced, fully autonomous and free-drifting buoy with technology to monitor and correct biases. Our chamber is small in size meaning that any turbulence passes the chamber without diminishing significantly at the oceanic regimes we deployed it, that has been proven by our continuous monitoring of turbulent kinematic energy under and outside the chamber's perimeter. We have shown in previous studies that the Turbulent Kinetic Energy (TKE) inside (TKE_{in}) and outside (TKE_{out}) were not statistically significant (Reference 10 and 29). It also proves that the chamber itself does not decrease turbulence as reviewer suggest. We would like to point out that we have described this in the original manuscript in lines 105-140. Overall, we disagree that our data are questionable simply based on the chosen technique. We have designed our chamber with much care over a two-year period undergoing various field tests and quality assurance procedure (see Ribas-Ribas et al., 2018). Furthermore, the chamber technique is the only applicable technique for our study to investigate small-scale variations of CO₂ air-sea transfer including a comparison between slick and non-slick areas. Neither Eddy covariance nor dual tracer (all with their own biases and challenges to apply on sea) can assess gas transfer velocity on sufficient small spatial and temporal scales to apply on slicks. In an another review process for our recent published paper in Geoscience (Ribas-Ribas et al. 2019) we received very positive feedback on our efforts to advance the chamber technique to produce high-quality data on gas transfer velocity. The reviewer stated that: “The technology (Sniffle) is promising and addressing some of the issues related to traditional chamber measurements and from its presentation it looks that it can provide new insights in parameterizations under low

wind conditions. The methodology and evaluation of the results is to a great extent robust.” (We can provide the original reviewer’s comment upon request)

Please refer to our publications using our floating chamber method:

Banko-Kubis, H. M., Wurl, O., Mustafa, N. I. H., and Ribas-Ribas, M.: Gas Transfer Velocities in Norwegian Fjords and the Adjacent North Atlantic Waters, *Oceanologia*, 61, <https://doi.org/10.1016/j.oceano.2019.04.002>, 2019.

Ribas-Ribas, M., Battaglia, G., Humphrey, M. P., and Wurl, O.: Impact of nonzero intercept gas transfer velocity parameterizations on global and regional ocean–atmosphere CO₂ fluxes, *Geosciences*, 9, <https://doi.org/10.3390/geosciences9050230>, 2019.

Ribas-Ribas, M., Kilcher, L. F., and Wurl, O.: *Sniffle*: a step forward to measure in situ CO₂ fluxes with the floating chamber technique, *Elem Sci Anth*, 6 (1): 14, <https://doi.org/10.1525/elementa.1275>, 2018.

There is also an issue that the transfer velocity data presented in the manuscript is very different from previous measurements. Below is a plot I generated of the data from Mustafa et al. (Figure 1) plotted against several recent field measurements of gas exchange (McGillis et al., 2004; Salter et al. 2011).

Thank you for the additional data compilation. It clearly shows the complexity of gas exchange and its measurement, despite the rather selective choice of data source by the reviewer. As pointed out in our manuscript, our data are similar with Donelan, M.A. & Drennan (1995) (Reference 40), which was ignored by the reviewer. We suggest that the high k₆₆₀ in our study, exclusively observed in the western Pacific (see Supplementary Fig. S2), is due to the low resistance of air–sea CO₂ transfer by lowest concentration of surfactants observed on a global scale (text line 198–201).

In addition, Salter’s et al. data under clean conditions and higher wind speeds (> 8 m s⁻¹) are occasionally as high as with the “gradient technique” at a wind speed of 3–5 m s⁻¹. We believe it is difficult to compare data obtained from different techniques without cross-validation. Also, Salter’s gas transfer velocity varies from about 8 cm h⁻¹ to 27 cm h⁻¹ at similar wind speeds, clearly indicating

It is clear that the Mustaffa et al. data is very much larger than the other datasets, which are in relatively good agreement. The authors do not even attempt to explain this huge increase in gas transfer velocity, but move on to study the effects of surfactants. I find this troubling since it appears there is an issue with their fundamental data, and to this reviewer anyway, it alone calls into question their conclusions.

that an apparent reversed mismatch to our observations has been found in the past, e.g., lower gas transfer velocities at higher winds speeds compared to our higher gas transfer velocities are lower wind speed. However, with our surfactant data in the SML, we can provide a reasonable mechanism for the occurrence of the high k_{660} , i.e. low resistance of air-sea CO₂ transfer due to very low surfactant concentrations in the western Pacific

The reviewer claims above that the floating chamber technique underestimate gas transfer, but here the reviewer realized, correctly, that our data provide a new upper range for the parameterization of gas transfer. It seems like that both main concerns raised by the reviewer contradict each other.

Due to long history of the wind-based parameterization based on Wanninkhof, it is likely a bias exists as we know (due to personal communications with the SOLAS community) that larger gas transfer velocities have been also measured by others, but omitted for publication as the parameterization has been set as the “true” reference. New technology, like our direct and proven floating chamber technique and catamaran S³, provides new insights into the complexity of the process. We need to consider that existing parameterization has been based initially on lake data, forced through a zero intercept and a 14C global budget, and later indirect measurements (mainly Eddy covariance and dual tracer) has shown a wide range of deviation (also shown by the reviewer’s plot).

Another problem arises from the reviewer’s comparison. If we look at the data from Salter et al. comparing oleyl alcohol and “clean” SML, a significant reduction is questionable due to the highly variable data. That is odd, as laboratory studies clearly showed a very strong reduction of oleyl alcohol as monolayer on gas exchange, which seems to be missing in Salter’s data. Anyway, oleyl alcohol will form a monolayer and cannot be comparable with natural SML or slick. We already mention this in the text line 297 - 300:

	“However, the insoluble properties of oleyl alcohol form a monolayer film that does not completely simulate a natural slick with its biofilm-like [7] and rheological properties [48], the latter through increased thickness (compared to non-slick SML or monolayers), and the presence of complex mixtures of soluble and insoluble surfactants [49].“
On a more metaphysical note, one thing I think would be good for the authors to explain is why they feel existing parameterizations need to be modified to account for surfactants. To my knowledge all of the commonly used parameterizations have been generated using field data. One thing that I think has been a true advancement in our understanding of the SML is that natural surfactants are ubiquitous in the ocean, seen everywhere anyone has measured them, with enrichment in the microlayer. The work of Frew et al. (2002, ref 15 in manuscript) shows that the effect of surfactants is seen at concentration levels well below the background surfactant concentration in the ocean. This implies that all the field measurements of gas exchange used to derive these parameterizations have included the effects of surfactants. It is not clear why an extra correction is required, and it would be good for the authors to discuss this effect.	We thank for the comment. Our manuscript leads to a new thinking in a way that k parameterizations cannot be applied in a generic approach simply due to the occurrence of two end members in surfactant concentrations, i.e. low-surfactant regions (like the western Pacific) and slicks with extremely high surfactant concentrations. Firstly, we highlight that slicks, a sea-surface phenomena of wave-damping, reduces the transfer of climate-relevant gases between the ocean and atmosphere. The phenomenon of slicks has been known for decades, but occurrence and dynamic behavior in size and shape has been an obstacle to assess its effect on air-sea interaction. We developed the technology and combine different field studies to describe the role of slicks in the field of air-sea interaction with the aim that the occurrence of slicks will be considered in future field studies and models. We have found that at a threshold of $200 \mu\text{g L}^{-1}$ significant reduction of k occurs, but a further reduction occurs at concentrations $1000 \mu\text{g L}^{-1}$ (see Figure 2a and Figure 3b). We show in this paper that slicks with surfactant concentrations above $1000 \mu\text{g L}^{-1}$ reduces gas transfer velocities k_{660} for CO_2 by 62% compared to ambient sea surfaces with concentrations of $200\text{-}600 \mu\text{g L}^{-1}$, i.e. a further significant reduction above from the threshold of $200 \mu\text{g L}^{-1}$. Reduction of k across slicks is the key finding of our manuscript, as clearly shown in Figure 4 in our manuscript. Considering that cyanobacterial slicks can, for example, occupy 20% of the Arabian Sea (Capone et al. 1997) implementation into Earth System Models of the effects by slicks is a necessary next step.

	Secondly, the western Pacific Ocean is generally another extreme end member, i.e., with very low surfactant concentrations. As stated in our manuscript, wind-based parameterization has been developed elsewhere with typical surfactant concentrations exceeding those in the western Pacific at least two-fold. That means, the application of the parameterizations to regions of the western Pacific, or other low-surfactant regions, leads to an error of at least 20%. For this reason, we disagree with the statement that effect of surfactants is seen below the background surfactant concentration in the ocean. At the time of Frew et. al (2002) surfactant concentrations in the SML were widely unknown (only occasional measurements were taken), and our study here provide the first surfactant data for the Pacific and identifying this region as low surfactant regime.
On a more technical note, the way the authors have plotted the data seems to done to obfuscate noise and issues with the data. (Figure 1 is an example of this, where previous gas exchange data are not shown). Figure 3 might be better if the k660 data were plotted as a function of wind speed, and then color coded as a function of surfactant concentration. Figure 5 might be more illustrative plotted in this manner as well. The benefit would be showing existing wind speed parameterizations of gas transfer velocity on the same plot would show the potential effect of including surfactants.	We did not intend to obfuscate noise and the data are available in PANGAEA data publisher as open access as outline in our manuscript (see references 59 and 60). In Figure 1, we intend to show the overall trend and for clarity we have binned the data, similar to McGillis et al. (2004). In Figure 3, we intend to show the breaking point at $200 \mu\text{g L}^{-1}$, and so we plotted against surfactants. Meanwhile in Figure 5, we intend to show the concentration of surfactant are influenced by different geographical location. Figure 5 is important for our claim that high transfer velocity occurred in the western pacific is due to low surfactant. However, we have added a new plot according to the reviewer suggestions to the supplementary material (see Supplementary Fig S2) References: [59] Ribas-Ribas, M. & Wurl, O. 2019 Measurements of pCO₂ and turbulence from an autonomous drifting buoy in 2016 during FALKOR cruise FK161010. PANGAEA. (doi:https://doi.org/10.1594/PANGAEA.897104). [60] Banko-Kubis, H., Wurl, O. & Ribas-Ribas, M. 2019 Measurements of pCO₂ and turbulence from an

	autonomous drifting buoy in July 2017 in the Norwegian fjords and adjacent North Atlantic waters during cruise HE491. PANGAEA. (doi:https://doi.pangaea.de/10.1594/PANGAEA.900728).
In summary, in general I am supportive of work looking to understand the impact of surfactants on gas transfer. However, it is not productive to make sweeping claims on the role of surfactants using data that might be questionable (see figure above). The authors need to provide a far more convincing case that the headspace method provides reliable estimates of the transfer velocity, and that the values are not biased by an interaction between the wind-stress free surface under the headspace and existing surfactants. I recommend the manuscript be returned to the authors to deal with these issues, and final decision made after review of the revised manuscript.	Thank you for your support to understand the impact of surfactant on gas transfer velocity. However, we disagree that our data are questionable simply based on the chosen technique. New techniques, or improving existing techniques, should not generally lead to the assumption that the data are questionable. We have published our technique and other data obtained with this technique, and, therefore, we don't understand why the reviewer request further convincing cases. Reviewer's data compilation does not justify to question our data as it shows also that low gas transfer velocity exists at high wind speeds (reverse from our observation).

Reviewer 2

The manuscript discusses the effect of surfactants in the sea-surface microlayer (SML) on gas transfer velocity (k) under natural conditions. An error of flux estimation in the western Pacific is induced by applying wind-based parameterization not developed in low surfactant regimes. Reduction in natural slicks reduces the global flux of CO_2 by 19%. Overall the study is very interesting and significant. The manuscript needs to be improved before it can be accepted for publication.

Comments	Response
I think the authors need to improve their abstract. What are the main objectives of this study? It is good if they can briefly mention the approach of their study.	We thank for the comment. Our manuscript aims to compare the reduction of gas transfer velocity (k) between non-slick and slick condition. The novelty of our study lies in the fact that we bridge decades-old laboratory studies to the real environment allowing the implementation of slicks to global carbon models; rather than using lab studies on artificial monolayers. Finally, we present a nearly 20% reduction of global CO_2 fluxes considering known frequency of slick coverage; representing a significant error if slicks are further ignored. Therefore, we revised our abstract by adding the main objective in line 18 – 23: “Moreover, a sea-surface phenomena of wave-damping, known as slicks, has been observed frequently in the ocean and potentially reduces the transfer of climate-relevant gases between the ocean and atmosphere. Therefore, this study aims to quantify the effect of natural surfactant and slicks on the in-situ k of CO_2. A catamaran, Sea Surface Scanner (S^3), was deployed to sample the SML and corresponding underlying water (ULW), and a drifting buoy with a floating chamber was deployed to measure the in-situ k of CO_2.”
The introduction is not well structured as well. The authors need to clearly mention the motivation for the study. What they	We thank the reviewer for the comment and we have improved paragraphing (in original

already knew and what they want to cover for this study. I suggest the authors improve their paragraphing in the Introduction.	manuscript line 26 - 55) of our introduction accordingly (now in line 33 – 67). The motivation of our study was to present the first in situ data on air-sea CO₂ transfer velocity in dependence to different levels of surfactants in surface films. In-situ measurements have been the most reasonable next step in understanding air-sea gas exchange processes as, and that is most critical, observations from the field are most credible to the implementation into global models. We also included literature on slick in our introduction (in text line 39-45). “Meanwhile, slick is a sea-surface phenomena of wave-damping effect by the excessive accumulation of organic matter. Slicks are frequently observed in the ocean [6] and potentially reduces the air-sea CO₂ exchange by 15% [7] based on data obtained from artificial monolayers. Natural SML and slicks have not been well explored in past research programs that estimate the fluxes of CO₂ into and out of the ocean. However, known bias of 20–50% in theoretical approaches [8, 9] controlled tank [10-12] and field experiments involving artificial SMLs [13] justify observation under natural conditions.”
Line 276: name the study that considers the effect of slick on K₆₆₀.	To our knowledge, there is no study considering the effect natural slick on the k₆₆₀. However, we already included the study considering the effect of artificial slick on k₆₆₀ in the text line 294-297: “For example, the field measurements using artificial slicks of oleyl alcohol, reducing micro-scaled turbulence under the surface by damping capillary waves, indicated suppression of k₆₆₀ up to 30% and 55% at low (1.5 – 3.0 m s⁻¹) [46] and high wind speeds (6.9 – 7.6 m s⁻¹) [11], respectively.”